# SELF-INFORMED GENERATIVE ACTIVE LEARNING

## ABSTRACT

Active learning has been a cost-efficient approach to obtaining high-performance AI models with fewer selective annotations. In scenarios where the acquisition of original unlabeled data poses significant challenges, active learning harnessing synthesized data instances is more promising than traditional pool-based methods. In this paper, we propose the Self-Informed Generative Active Learning (SIGnAL) framework as an effective solution to actively generate and select data instances for annotation and downstream model training. In SIGnAL, we propose to guide the data generation based on a reinforcement learning policy, where the generator is self-informed by the reward to generate more informative instances. In addition, we introduce an acquisition function that measures both the informativeness and relevance of instances. Such acquisition function can be transformed to the reward seamlessly for generator optimization. Our experiments on the text classification task validate the effectiveness of our framework, especially when the original data scale is limited.

## 1 INTRODUCTION

Active learning has been an effective solution to the contradiction between the demand for supervised training data and the high cost of data annotation. High-quality data, especially data with annotations or human feedback, is crucial for the performance of various AI models. Empirical analysis has shown that even SOTA models are significantly undertrained and can benefit from more data (Kaplan et al., 2020; Hoffmann et al., 2022). However, constructing diverse, high-quality datasets poses challenges, as manually annotating data can be both costly and time-intensive. Toward this end, active learning seeks to select from the data pool fewer but more informative data instances, of which annotations are obtained to add significant improvements to the trained models.

Existing active learning research centers on pool-based approaches, which aim to select the most informative instance or batch from a pool of unlabeled data. However, pool-based methods face two main limitations: First, instances from the unlabeled pool might not be optimal across the entire data space. For example, techniques like regularization (DeVries, 2017; Geirhos et al., 2018; Sun et al., 2020) and adversarial perturbation (Miyato et al., 2016) create out-of-distribution (OOD) data that lead to better model generalization and robustness. Second, pool-based methods assume that a large unlabeled pool of data is readily accessible. In situations where collecting unlabeled data itself is costly or challenging, such as in the robotics and medical fields, the available data tend to suffer from biased distributions or be insufficient for training.

These limitations motivate query-synthesizing approaches to use generative models to produce informative instances with higher informativeness. In theory, generative models allow broader exploration of the data space and reduce dependence on a large unlabeled pool. However, such exploration turns out to be challenging in practice for two reasons: First, optimizing the generative model requires translating the concept of informativeness into learnable signals, but informativeness is difficult to be defined as a simple loss function due to the absence of a definitive correct solution and its inherently delayed nature. Specifically, we cannot directly provide the most informative data point for the model to learn from, and informativeness is assessed at the end of the progressive steps of generation. Second, the over-optimization and randomness feature of the generative model can lead to the produce of irrelevant OOD data (e.g., $q$ in Figure 1). Although might be considered informative following traditional definitions such as uncertainty-based methods, this type of instance is irrelevant to the original data distribution and can negatively impact model performance.

To address the mentioned challenges in query-synthesizing active learning, in this paper, we propose the Self-Informed Generative Active Learning (SIGnAL) framework (Figure 2). SIGnAL is a general reinforcement learning (RL)-based framework for both data instance generation and selection in active learning. The contributions of this work are as follows. First, we provide a smooth solution to the challenge of the dynamic and delayed nature of informativeness through the usage of reinforcement learning. Specifically, we serve instance informativeness as the reward signal from the concept of reinforcement learning to optimize the generative model, since reinforcement learning is well-suited for handling dynamic optimization targets and unclear ordinary losses. Second, we design an acquisition function that evaluates both the traditional informativeness and the relevance of data instances, which is transformed into the reward during training. Third, we provide a practical method to align large language models (LLM) with our RL-based framework and validate its effectiveness on text classification tasks. It is worth mentioning that SIGnAL is a general framework that has the potential of leveraging the advantages of various SOTA large generative models in different tasks.

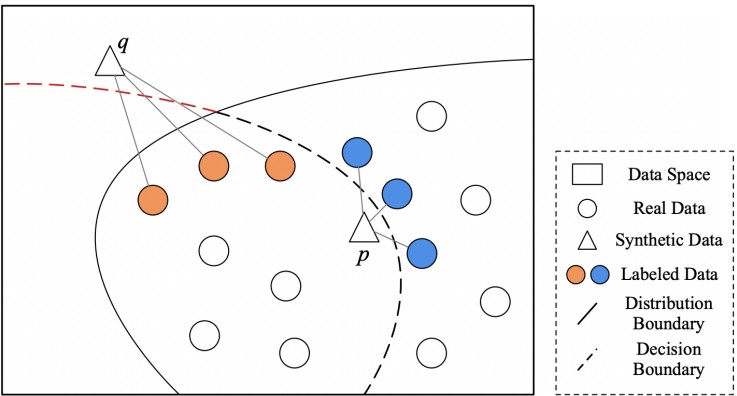

Figure 1: An illustrative example for our proposed acquisition function. Two synthetic data points $p, q$ are both highly contrastive with their nearest neighbours, but the acquisition function deems $p$ as more informative since it is closer to its neighbours.

## 2 RELATED WORKS

### 2.1 ACTIVE LEARNING

Current active learning approaches can be categorized as pool-based or query-synthesizing. Pool-based algorithms utilize different sampling strategies to select the most informative data points from the unlabeled set, while query-synthesizing methods leverage generative models to synthesize informative samples. Our work aligns with the latter, a less explored but increasingly significant field as the capabilities of generative models advance.

Pool-based algorithms can be further broken down into uncertainty-based methods, diversity-based methods, and mixed methods. Uncertainty-based approaches quantify the uncertainty of data points and select those with high uncertainty. Simple definitions of uncertainty (Nguyen & Smeulders, 2004) include confidence (Lewis & Catlett, 1994), margin (Joshi et al., 2009), and entropy (Shannon, 1948; Luo et al., 2013) that make use of model posterior probabilities. Recent advancement in uncertainty measurement include using Monte Carlo dropout (Gal et al., 2017) for deep neural networks and attaching a loss prediction module that estimates the loss of unlabeled data (Yoo & Kweon, 2019). In contrast, diversity-based methods select batches of data points representative of the unlabeled set. Clustering methods have been widely applied to choose data points closest to the cluster centroids. The core-set approach (Sener & Savarese, 2017) marks a pivotal advancement in diversity-based methods. It frames active learning as a core-set selection problem, which aims to find a subset of the full dataset such that the model trained on the subset effectively approximates the whole dataset. Mixed approaches seek to leverage the advantages of both uncertainty and diversity. For example, Batch Active learning by Diverse Gradient Embeddings (BADGE) (Ash et al.,

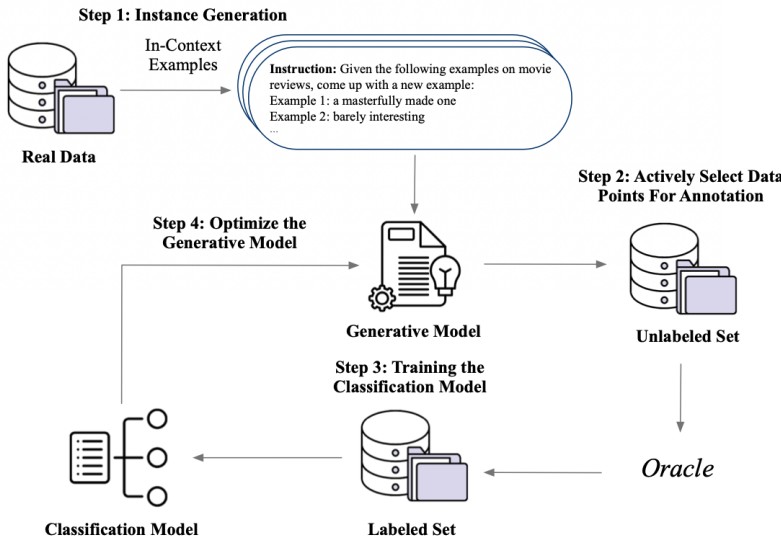

Figure 2: A high-level overview of SIGnAL. The workflow begins with the generative model producing unlabeled instances based on in-context examples. These instances are added to the unlabeled set, where the classification model actively selects data points for annotation and is then retrained on the expanded labeled set. Then, the classification model assesses the informativeness of the generated instances, providing a reward signal to optimize the generative model for subsequent iterations.

2019) selects data points by clustering their hypothetical gradient embeddings, and Contrastive Active Learning (CAL) (Margatina et al., 2021) defines informative data points as those contrastive with their neighbors in the data space.

Another line of research is query-synthesizing methods, which seek to utilize generative models to create informative data points. Generative Adversarial Active Learning (GAAL) (Zhu & Bento, 2017) is the first work that incorporates deep generative models into solving the active learning problem. It employs a GAN to generate informative data points through descending the gradient of a simple loss function. However, GAAL reports lower accuracy than the random sampling baseline, because the generative model is over-optimized and produces irrelevant and indistinguishable instances in later stages of training. Subsequent works, including Adversarial Sampling for Active Learning (ASAL) (Mayer & Timofte, 2020)and the Bayesian Generative Active Deep Learning (BGADL) (Tran et al., 2019), use a mixed dataset of synthetic and real data. ASAL generates synthetic instances and then identifies similar real data within the unlabeled pool, while BGADL initially queries the unlabeled set and then synthesizes similar data points. Nevertheless, these methods are based on similarity instead of looking for potentially more informative data points.

## 2.2 SYNTHETIC DATA GENERATION

Collecting large-scale human-written data is time-consuming and labor-intensive. As powerful instruction-tuned large language models emerge, LLM-generated data have been widely used in model training and shown comparative performance to human-written data. The Self-Instruct framework (Wang et al., 2022) introduces an automated pipeline for generating high-quality instruction-tuning data with only LLMs. It conditions data generation on a set of 175 seed tasks and applies filtering to obtain high-quality data. and conditions data generation on the tasks Finetuned on the 52k instruction-response pairs generated by OpenAI's Text-Davinci-003, the Alpaca model (Taori et al., 2023) performs on pair with Text-Davinci-003, which is trained with private user data and human annotations. More recently, Microsoft's Phi series (Gunasekar et al., 2023; Li et al., 2023; Abdin et al., 2024) has incorporated billions of tokens of textbook-quality synthetic data into its training process, further demonstrating the effectiveness of synthetic data.

## 2.3 ALIGNMENT WITH REINFORCEMENT LEARNING

The modeling objective for large language models – predicting the next token on a large corpus of text from the Internet – often diverges from the objective of following the user's instructions helpfully and safely. Yet "helpful and safe" is an abstract concept and is difficult to encode into a loss function in the supervised setting since the problem has no single correct answer and involves sequence decision-making. Therefore, reinforcement learning has been applied to align LMs in various NLP tasks, ranging from text summarization (Ziegler et al., 2019) to story generation (Zhou & Xu, 2020). Building on these efforts, Ouyang et al. proposes the RLHF framework (Ouyang et al., 2022), which first trains a reward model based on human preferences, and then finetunes the LM using reinforcement learning. The success of the resulting model, InstructGPT, has established reinforcement learning as a paradigm for aligning LMs with abstract, indirect objectives.

## 3 PROBLEM DEFINITION

In this section, we formally define the generative active learning problem and set up the notations for the rest of the paper. With a target classification model $M(x; \theta)$ and a generative model $G(x; \phi)$, we consider a $C$ classification problem defined over a data space $\mathcal{X}$ to a label space $\mathcal{Y} = \{1, ..., C\}$. We also consider a loss function $l(\cdot, \cdot) : \mathcal{X} \times \mathcal{Y} \to \mathbb{R}$.

We assume an underlying data distribution $p_z$, and we have access to a pool of unlabeled data points $\{x_i\}_{i \in [n]}$ drawn i.i.d. from $p_z$, which form the real data pool $\mathcal{R}$. In addition, we can use generative model to create extra data points $\{x'_i\}_{i \in [m]} = G(x; \phi)$, which form the synthetic data pool $\mathcal{S}$. Together, $\mathcal{R}$ and $\mathcal{S}$ form the unlabeled pool $\mathcal{U}$.

Given an annotation budget $b$, the generative active learning problem can be defined as:

$$\min_{s \subseteq \mathcal{U}, |s| \leq b} E_{(x,y) \sim p_z}[l(M(x; \theta_s), y)]$$

where $\theta_s$ is a set of parameters learned on a labeled set $s$ by minimizing $E_{(x,y) \sim s}[l(M(x; \theta), y]$

The active learning problem is the same as in the pool-based setting with two exceptions: First, we introduce a new synthetic unlabeled pool $\mathcal{S}$ in addition to the real data pool $\mathcal{R}$. Second, we also consider the case where $n$ is small; for example, $n < b$.

## 4 METHOD

In this section, we introduce the proposed SIGnAL method. We start with the generation of new unlabeled data using a language model in section 4.1. Next, we define the acquisition function that serve both as the sampling criterion and reward signal in section 4.2. Finally, we describe the optimization of the generative model with the feedback from the classification model in section 4.3.

### 4.1 DATA GENERATION

In this work, we use pretrained large language models for text data generation. A principal challenge of LLM-generated data is that they tend to be repetitive and redundant as LLMs tend to follow the most probable paths based on their priors. Filtering synthetic data based on similarities to existing data is one possible solution, but since generation is expensive, we employ another strategy of crafting a diverse set of prompts to promote diversity in the generated texts.

For each generation, the prompt comprises an instruction $I$ that specifies the domain of the dataset and the task and $p$ in-context examples. The in-context examples are chosen at random from the real distribution to promote the generation of in-distribution data. The generation process can be characterized by

$$x' = \arg\max_x P(x|I, x_1, ..., x_p; \phi)$$

Here is an example generation from the SST-2 (Socher et al., 2013) dataset:

**Prompt**: Based on the following examples of movie reviews, come up with a new example: Example 1: of an authentic feel

---

**Algorithm 1** Single Iteration of SIGnAL

---

**Require:** Classification model $M(x; \theta)$, generative model $G(x; \phi)$, real dataset $\mathcal{R}$, unlabeled dataset $\mathcal{U}$, labeled dataset $\mathcal{L}$, prompt size $p$, annotation budget $b$
1:  Randomly choose $p$ in-context examples $\{x_1, ..., x_p\}$ from $\mathcal{R}$
2:  Generate $b$ synthetic instances $\{x_i'\}_{i \in [b]} \leftarrow G(I, x_1, ..., x_p; \phi)$
3:  Update $\mathcal{U} \leftarrow \mathcal{U} \cup \{x_i'\}_{i \in [b]}$
4:  **for** $x_i \in \mathcal{U}$ **do**
5:      Find $k$ nearest neighbors in the labeled set $\{x_l^{(j)}\}_{j \in [k]} \leftarrow \text{KNN}(\Phi(x_u), \Phi(\mathcal{L}), k)$
6:      Compute informativeness score:

$$s_{x_i} \leftarrow \frac{1}{k} \sum_{j=1}^{k} \frac{\text{KL}(p(y|x_i) \parallel p(y|x_l^{(j)}))}{d(\Phi(x_i), \Phi(x_l^{(j)}))}$$

7:  **end for**
8:  Select batch $B \leftarrow \arg\max_x s_x$, $|B| = b$
9:  Update $\mathcal{L} \leftarrow \mathcal{L} \cup B$
10: Update $M$: $\theta^* \leftarrow \arg\min_\theta E_{(x,y) \sim \mathcal{L}}[l(M(x; \theta), y)]$
11: Construct RL dataset $D_{RL} \leftarrow \{(I, x_1, ..., x_p)_i, x_i'\}_{i \in [b]}$
12: Define reward $r((I, x_1, ..., x_p), x') \leftarrow s_{x'}$
13: Update $G$: $\phi^* \leftarrow \arg\max_\phi E_{(x,y) \sim D_{RL}} \left[ r(x, y) - \beta \log \left( \pi_\phi^{\text{RL}}(y \mid x) / \pi^{\text{Pretrained}}(y \mid x) \right) \right]$

---

    Example 2: that really , really , really good things can come in enormous packages
    Example 3: is listless , witless , and devoid of anything
    New Example:
    **Generated Instance**: quick-paced, witty exploration of contemporary society

We repeat this procedure to generate $b$ synthetic data and add them to the unlabeled set.

## 4.2 ACQUISITION FUNCTION

Pool-based acquisition functions are designed under the assumption that the sample distribution represents the underlying data distribution. However, this assumption no longer holds with the presence of synthetic data, which may come from a completely different data distribution. Thus, we introduce a relevance metric to informativness measurement in the generative setting.

First, we define relevance as the closeness between the model encodings of a data point and its neighbors. A data point $x_i$ is irrelevant if the distance between the model encodings of itself and its nearest neighbor $x_j$ maximally diverge:

$$d(\Phi(x_i), \Phi(x_j)) \to \infty$$

Next, we follow the definition of informativeness in the work CAL (Margatina et al., 2021). A data point $x_i$ is informative if the predicted likelihood between itself and its nearest neighbor $x_j$ maximally diverge:

$$\text{KL}(p(y|x_i) || p(y|x_j)) \to \infty$$

Combining these two definitions, we get the following acquisition function:

$$s_{x_i} = \frac{1}{k} \sum_{j=1}^{k} \frac{\text{KL}(p(y|x_i) || p(y|x_j))}{d(\Phi(x_i), \Phi(x_j))}$$

where $\{x_j\}_{j \in [k]}$ are the $k$ nearest neighbors for $x_i$. This acquisition function aims to find data points whose predictive likelihood is contrastive with those of their neighbors while staying close to the distribution.

Table 1: Dataset statistics

| DATASET | TASK | TRAIN | VAL | TEST | CLASSES |
|---------|------|-------|-----|------|---------|
| SST-2 | Sentiment Analysis | 67.3K | 872 | 1.82K | 2 |
| AGNEWS | Topic Classification | 120K | - | 7.6K | 4 |
| QNLI | Natural Language Inference | 105K | 5.46K | 5.46K | 2 |

### 4.3 GENERATOR OPTIMIZATION

In this section, we aim to optimize the generative model towards producing more informative data points. We achieve this using reinforcement learning and the PPO algorithm (Schulman et al., 2017).

First, we construct a dataset consisting of prompt-response pairs for reinforcement learning $D_{RL} = \{(I, x_1, ..., x_p)_i, x'_i\}_{i \in [b]}$. Next, we define the reward for each prompt-response pair as the informativeness score of thegenerated instance

$$r((I, x_1, ..., x_p), x') = s_{x'}$$

Then, we use the PPO algorithm to finetune a new RL policy. Specifically, we maximize the following combined objective function in RL training:

$$\text{objective}(\phi) = E_{(x,y) \sim D_{RL}} \left[ r(x, y) - \beta \log \left( \pi_\phi^{\text{RL}}(y \mid x) / \pi^{\text{Pretrained}}(y \mid x) \right) \right]$$

where $\pi_\phi^{\text{RL}}$ is the learned RL policy, and $\pi^{\text{Pretrained}}$ is the pretrained policy. The KL penalty from the pretrained model mitigates over-optimization of the reward model.

## 5 EXPERIMENT

### 5.1 TASKS AND DATASETS

We evaluate SIGnAL across multiple text classification tasks. Specifically, we use SST-2 (Socher et al., 2013) for sentiment analysis, AGNEWS (Zhang et al., 2015) for topic classification, and QNLI (Wang, 2018) for natural language inference. To simulate scenarios with a limited initial unlabeled pool, we randomly sample $0.1\%$ and $1\%$ of the original size from each dataset. Since the SST-2 and QNLI are a part of the GLUE benchmark and the labels for their test sets are not publicly available, we use their respective validation sets for evaluation.

### 5.2 BASELINES

We compare SIGnAL with five pool-based baselines, as existing query-synthesizing methods are designed to handle image data. Random functions as a baseline where no active learning is involved. Entropy (Luo et al., 2013) is the most commonly used uncertainty-based baseline method and selects points with the highest predictive entropy. BERTKM (Yuan et al., 2020) is a diversity-based baseline which applies applies k-means clustering using the $l_2$ normalized BERT embeddings and chooses the nearest data point to each center. BADGE (Ash et al., 2019) and CAL (Margatina et al., 2021) are two recently proposed methods that generalize the uncertainty and diversity principles. Specifically, BADGE computes the hypothetical gradient embeddings of each data point and clusters them with k-means++. CAL chooses data points whose predictive probabilities differ from those of their neighbors.

### 5.3 IMPLEMENTATION DETAILS

We use BERT-BASE (Devlin, 2018) with a task-specific classification layer as the target classification model and Qwen2.5-7B-Instruct (Yang et al., 2024) as the generative model. To assess model performance, we measure accuracy at increments of an additional $10\%$ of labeled data. While traditional pool-based methods halt once $100\%$ of the data has been acquired, SIGnAL continues by actively generating new data throughout the process. Nonetheless, we limit the evaluation of SIGnAL's accuracy to up to $200\%$ of the acquired data, which is sufficient for understanding its behavior.

Each active learning method is evaluated multiple times with varying initial labeled datasets, and we report the mean and standard deviation of the performance across trials. To ensure a fair comparison, the same random seed is used across all methods for each trial.

A key challenge in experiments of active learning with synthetic data is annotation. In pool-based experiments, all labels are known but remain hidden from the model until selected for annotation. However, the labels for synthetic instances are not known in advance. To address this, we use classification models fine-tuned on the respective datasets to annotate the synthetic instances. The annotators achieve accuracies of 91.3% on SST-2, 93.75% on AGNEWS, and 90.99% on QNLI. Although we assume the oracle provides 100% accurate annotation, this could result in performance degradation in practice.

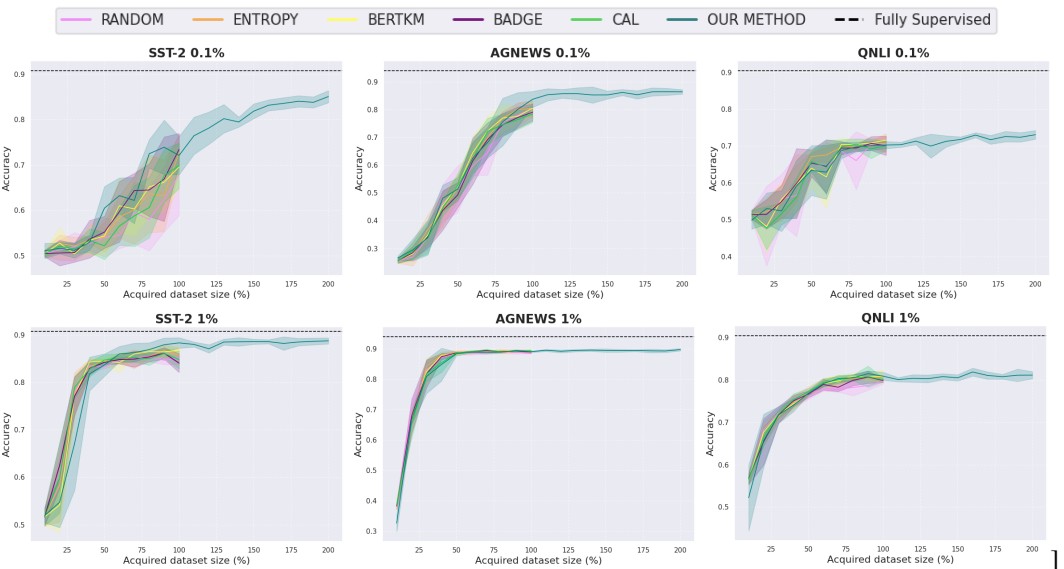

Figure 3: Test accuracy for different AL methods under different acquired dataset size

## 5.4 ANALYSIS

We present the results of our experiments in 3. We observe that SIGnAL consistently outperforms pool-based methods across all datasets and dataset sizes, especially in SST-2 0.1%, SST-2 1%, and AGNEWS 0.1%.

When comparing different dataset sizes, SIGnAL tends to perform better on smaller datasets. This is due to two key reasons. First, smaller datasets have a higher likelihood of being biased, as the limited data may not adequately represent the underlying distribution. In the generative setting, the model can learn to explore underrepresented areas of the data distribution, thereby mitigating bias. Second, as the dataset size increases, the model experiences diminishing returns: while larger datasets continue to improve performance, the rate of improvement gradually decreases.

When comparing different datasets, we observe that SIGnAL tends to perform better on SST-2 and AGNEWS than on QNLI. This can be explained by the similarity between the data distribution of the task and the training distribution of the generative model. SST-2 consists of movie reviews that are either positive or negative, and AGNEWS contains new articles across categories of world, sports, business, and sci/tech. These data are prevalent on the Internet, from which the training corpus of the generative model is sourced. However, QNLI contains question-response pairs where the response either contains the answer to the question or not. The generative model is less probable to create responses that do not contain the answer to the question, which leads to a bias towards entailment (response containing the answer to the question) data. Nevertheless, we observe that while the generative model starts to create exclusively entailment data, it gradually learns to produce more not-entailment data as entailment data become less informative to the classification model. This

adaptive behavior explains why SIGnAL performs bette on QNLI $1\%$ as compared to QNLI $0.1\%$: The generative model requires time to adjust its generation policy towards producing more balanced labels.

In each experiment, we observe a consistent pattern: SIGnAL tends to underperform compared to pool-based methods during the early stages of training. A closer examination of the generated data reveals that, in the beginning, the generative model often produces instances similar to the in-context examples, which can lead to inefficient use of the annotation budget if these repetitive instances are selected. However, as training progresses, the generative model gradually learns what constitutes informative data and begins to generate instances that are more informative than real data. This observation suggests a potential improvement for SIGnAL: implementing an adaptive budget allocation strategy that progressively shifts from relying on real data to incorporating more synthetic data.

Lastly, we compare the performance of different pool-based acquisition functions. Overall, all methods outperform the random baseline. Among them, BERTKM achieves the best performance in most experiments, highlighting the effectiveness of diversity-based sampling on smaller datasets. Additionally, ENTROPY ranks among the top two performing acquisition functions across all experiments despite its simplicity. Two combined methods, BADGE and CAL, deliver more mediocre results in our experimental setting.

## 6   CONCLUSION

In this work, we introduce SIGnAL, an RL-based query-synthesizing framework that actively generates and selects data instances for annotation and downstream model training. We also propose an AL acquisition function for a pool made up of both real and synthetic data. We have demonstrated the effectiveness of SIGnAL on multiple text classification tasks with a limited unlabeled pool.

SIGnAL constitutes an initial effort that integrates LLMs into the traditional AL paradigm, which implies that future works can explore improvements to SIGnAL from multiple angles. Future authors can follow the traditional line of AL research and design more effective acquisition functions, or explore more effective or computationally efficient ways of optimizing the generative model. Further, while we apply SIGnAL specifically to text data in this work, it is flexible framework that can be applied to other forms of data, such as images.

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
