# OpenReview forum: "Self-Informed Generative Active Learning"
_ICLR.cc/2025/Conference — Submitted to ICLR 2025_

### Official Review · Reviewer_JtSL · 2024-10-29

**Soundness:** 2
**Presentation:** 2
**Contribution:** 2
**Rating:** 3
**Confidence:** 4

**Summary:**

This paper introduces an active learning approach for NLP tasks utilizing a generative model. It incorporates KL divergence, as proposed in CAL, to retrieve informative samples and uses inter-sample distance to avoid querying unrelated samples. The method outperforms comparable approaches and can continue the active learning process even without access to further unlabeled data by leveraging generated samples. However, the limited datasets and reliance on LLM raise questions about the necessity of the approach.

**Strengths:**

1. The proposed approach outperforms other techniques, such as CAL, BERTKIM, and BADGE.
2. It allows performance gains by querying generated data after exhausting unlabeled data.

**Weaknesses:**

1. The utility of this approach is ambiguous. Active learning aims to efficiently query valuable samples in low-data regimes, particularly in areas with difficult labeling requirements, such as medical or legal fields.:
- 1.1. The paper only presents general datasets (SST-2, AGNEWS, QNLI) focused on tasks like sentiment analysis and topic classification, where active learning might be unnecessary. Given that the LLM itself can achieve higher performance on such tasks, training an additional classifier via active learning seems contradictory. For this approach to be useful, the active learning-trained model should outperform the LLM.
- 1.2. In this regard, domain-specific datasets, such as PubMed or legal datasets should be added. However, studies suggest that even specific tasks can achieve performance gains without active learning (or human labeling) through LLMs [1], raising questions about this method's utility compared to such approaches.
2. The number of datasets and class diversity are limited, with only three datasets and two or four classes per dataset. Include datasets with more classes, like DBPEDIA with 14 classes, to address whether the proposed method benefits persist as class counts increase.
3. The main paper lacks a definition of $\Phi$, which can only be inferred as a text encoder model.
4. No ethics statement is provided. An ethics statement and societal impact are mandatory for ICLR.
5. Hyperparameters are not disclosed. Without code submission, at least hyperparameter settings or a code statement should be included.
6. Time consumption details are missing. Given the method's reliance on LLMs and RL and the continuous dataset expansion, it likely requires considerably more time than alternative methods. Please add this information.

[1] Kim et al., "SELF-EXPERTISE: Knowledge-based Instruction Dataset Augmentation for a Legal Expert Language Model"

**Questions:**

1. What is the ratio of generated samples to actual unlabeled data queried at each iteration in the active learning process? If generated samples are only queried after unlabeled data, their value seems minimal.
2. How were the generated samples human-labeled? It’s likely some generated samples are incoherent, making labeling challenging. The paper includes no information about the human labeling process.

---

### Official Review · Reviewer_VPBQ · 2024-10-31

**Soundness:** 2
**Presentation:** 3
**Contribution:** 3
**Rating:** 3
**Confidence:** 4

**Summary:**

This paper leverages a RL policy to guide data generation, allowing the generator to receive rewards that encourage the creation of more informative instances. Additionally, it introduces an acquisition function that evaluates both informativeness and relevance, seamlessly transforming this evaluation into rewards for optimizing the generator.

**Strengths:**

By utilizing reinforcement learning, the approach effectively addresses the challenges posed by the dynamic and delayed nature of informativeness, treating instance informativeness as the reward signal to optimize the generative model. The method incorporates an acquisition function that evaluates both traditional informativeness and the relevance of data instances, transforming these evaluations into rewards during training.

**Weaknesses:**

The paper provides a detailed analysis of the challenges faced by pool-based active learning methods; however, it lacks an introduction to existing query-synthesizing methods and a distinction between the proposed method and existing synthesizing-based methods, such as “LADA: Look-Ahead Data Acquisition via Augmentation for Deep Active Learning” and “When Active Learning Meets Implicit Semantic Data Augmentation”. However, synthesizing-based methods are one of the primary categories in the active learning scenarios.

The PPO reinforcement learning method is utilized in this paper to optimize active learning strategies for larger rewards. Could you provide a detailed explanation of the state and action settings in this reinforcement learning scenario? Additionally, is it worth considering adopting the classifier's accuracy as an additional reward after generating samples?

Regarding the experiments:  The baseline methods adopted in the paper are all pool-based active learning methods. To further validate the effectiveness of your method, it is suggested to compare with synthesizing-based methods as well. Moreover, according to the experimental setup, synthesizing-based methods annotated twice as much more data, which could account for their superior performance. It is recommended to include ablation studies to provide additional explanations.

**Questions:**

The PPO reinforcement learning method is utilized in this paper to optimize AL strategies for larger rewards. Could you provide a detailed explanation of the state and action settings in this RL scenario?

Additionally, when considering other RL-based active learning strategies, is it worth considering adopting the classifier's accuracy as an additional reward after generating samples?

---

> ### Comment · Reviewer_VPBQ · 2024-11-26
>
> As the authors do not provide any feedbacks, I would like to decrease my rating.

---

> > ### Comment · Reviewer_aZDs · 2024-11-26
> >
> > Reviewer VPBQ, I don't think this (you decreasing the original score simply because the authors did not provide a rebuttal) is fair, especially when the rebuttal period is not done yet. It is doing nothing good.

---

### Official Review · Reviewer_NgKa · 2024-11-01

**Soundness:** 1
**Presentation:** 2
**Contribution:** 2
**Rating:** 3
**Confidence:** 4

**Summary:**

The paper proposes the Self-Informed Generative Active Learning (SIGnAL) framework, which generates synthetic data to improve active learning when real data is scarce. Using reinforcement learning, SIGnAL’s generator produces informative data guided by a reward system, ensuring relevance and usefulness for model training. An acquisition function assesses this data’s informativeness and relevance, optimizing the generator’s outputs. Experiments on text classification validate SIGnAL’s effectiveness, particularly in data-limited scenarios, offering a cost-efficient solution.

**Strengths:**

The idea is interesting and novel, the paper is easy to follow.

**Weaknesses:**

1. The experiments are far from sufficient for a top-tier conference, now there is only overall performance but lack of ablation study and analysis.
2. As a method that combines active learning and synthetic data generation from LLM, the authors only compare it with active learning approaches, I think they should also compare the proposed method with synthetic data generation without active learning

Missing related work:

[1] Large Language Model as Attributed Training Data Generator: A Tale of Diversity and Bias
[2] ZeroGen: Efficient Zero-shot Learning via Dataset Generation
[3] Generating Training Data with Language Models: Towards Zero-Shot Language Understanding

**Questions:**

In the equation of line 186, the distribution shouldn't be p_z because there is synthetic data while p_z is defined as real data, right?
Line 284: missing space between the and generate

---

### Official Review · Reviewer_aZDs · 2024-11-02

**Soundness:** 2
**Presentation:** 3
**Contribution:** 2
**Rating:** 3
**Confidence:** 4

**Summary:**

The paper introduces Self-Informed Generative Active Learning (SIGnAL), a RL-based approach for query-synthesizing active learning. SIGnAL generates synthetic data instances to enrich the data pool, especially when access to diverse, unlabeled real data is limited. Experimental results show SIGnAL’s performance advantage over traditional pool-based methods in text classification tasks, particularly when the data pool is very small.

**Strengths:**

1. The method addresses the limitations of traditional pool-based methods by generating informative synthetic data instances, this could be beneficial when even unlabeled data is scarce.
2. The paper is mostly well-organized.
3. The acquisition function that combines both informativeness and relevance makes sense.

**Weaknesses:**

1. The proposed SIGnAL does not generate the most informative/beneficial data point for labeling, instead, it still requires traditional acquisition function to make the selection. I think this is a critical weakness to this paper. From my understanding, generative AL should not only generate data samples, but more importantly generate the most informative samples.
2. The settings of this paper is king of niche, most areas that benefit from AL have abundant amount of unlabeled data, if SIGnAL simply generates more unlabeled data, I don't see it being very useful in practice.
3. The acquisition (relevance and informativeness) is quite simple, relevance is simply the distance, with informativeness directly taken from CAL.
4. The experiments are very limited. The only results are in Figure 3, with limited datasets, baselines, and the improvements are hardly distinguishable in my opinion.

In general I think this paper presents an interesting direction, but the details needs a bit more refinements.

**Questions:**

As discussed in the weakness section.

---

### Official Review · Reviewer_eSUZ · 2024-11-06

**Soundness:** 3
**Presentation:** 2
**Contribution:** 2
**Rating:** 3
**Confidence:** 3

**Summary:**

This paper addresses active learning by leveraging a generative model to produce unlabeled examples, which are then labeled by an oracle and added to a classification model's training set. Unlike traditional methods that rely on a fixed pool of unlabeled data, this approach actively generates new, potentially more informative examples. The model prioritizes examples based on their distance from nearest neighbors and the discrepancy in predictions between the generated sample and its neighbors. To guide the generative model in producing high-quality samples, it is trained via a Reinforcement Learning algorithm (PPO), optimizing it to generate samples that best serve the classification task. The method is tested on text classification problems, showing mixed results compared to current state-of-the-art techniques.

**Strengths:**

The approach is promising; using a generator to produce new samples is a valuable innovation for improving active learning systems. This strategy assumes a pre-trained generative model, which is reasonable for text but may not be universal across domains. The selection criterion is sensible, and directly training the generator to maximize it through RL is more robust than simple thresholding.

**Weaknesses:**

However, the experimental section lacks detail to fully evaluate the approach. Key hyperparameters—such as the number of samples generated per iteration and PPO settings—are not systematically analyzed, and no ablation study is provided. It would also be valuable to see a comparison of results with and without the RL approach. The current experimental section leaves significant space unexplored, making it hard to discern the model’s strengths and weaknesses.

It’s also unclear how the RL component is applied: Is the policy trained concurrently with sample generation, or is it established before the active learning phase? If the reward function evolves as new samples are generated, this could introduce non-stationarity, which would impact performance. Further clarification on this point is essential.

Regarding performance, the results do not clearly outperform existing methods. Notably, the learning curves for the proposed method (Signal) appear to extend longer than others. This might be because other methods are restricted to samples in the original dataset, while Signal can generate an infinite number of examples. However, this is not entirely clear, as baseline methods don’t achieve fully supervised performance, which raises questions about their comparison criteria.

**Questions:**

In conclusion, the paper presents an interesting idea, but the experimental section needs significant refinement. Adding more comprehensive experiments and ablation studies would strengthen the conclusions and clarify the potential of this approach.

---

### Meta-Review · Area_Chair_CwNT · 2024-12-11

**Metareview:**

This paper proposes a new method for active learning called Self-Informed Generative Active Learning (SIGNAL). The method uses a generative model to produce new examples, which are then labeled and added to the training set. This allows the model to actively generate new and potentially more informative examples. The model prioritizes examples based on their distance from nearest neighbors and the discrepancy in predictions between the generated sample and its neighbors. A reinforcement learning algorithm is used to train the generative model to produce high-quality samples. The method is tested on text classification problems.

(b) Strengths of the paper
The approach is promising.

Using a generative model to produce new samples is a valuable innovation for improving active learning systems.

The selection criterion is sensible.

Directly training the generator through reinforcement learning is more robust than simple thresholding.

The paper is well-organized.

The acquisition function that combines both informativeness and relevance makes sense.

(c) Weaknesses of the paper
The experimental section lacks detail and does not fully evaluate the approach.

Key hyperparameters are not systematically analyzed and no ablation study is provided.

A comparison of results with and without the reinforcement learning approach would be valuable.

It is unclear how the reinforcement learning component is applied.

The results do not clearly outperform existing methods.

The learning curves for the proposed method appear to extend longer than others.

The proposed SIGNAL does not generate the most informative data point for labeling and still requires a traditional acquisition function to make the selection.

The settings of the paper are a niche.

Most areas that benefit from active learning have an abundant amount of unlabeled data.

The acquisition is quite simple.

Relevance is simply the distance, with informativeness directly taken from CAL.

The experiments are very limited.

The only results are in Figure 3, with limited datasets and baselines, and the improvements are hardly distinguishable.

The experiments are far from sufficient for a top-tier conference.

There is only overall performance but a lack of ablation study and analysis.

As a method that combines active learning and synthetic data generation from LLM, the authors only compare it with active learning approaches.

They should also compare the proposed method with synthetic data generation without active learning.

(d) Reasons for rejecting the paper
The paper presents an interesting idea, but the experimental section needs significant refinement. Adding more comprehensive experiments and ablation studies would strengthen the conclusions and clarify the potential of this approach. The proposed SIGNAL does not generate the most informative data point for labeling, instead, it still requires a traditional acquisition function to make the selection. The settings of the paper are a niche, and most areas that benefit from active learning have an abundant amount of unlabeled data. The experiments are far from sufficient for a top-tier conference; there is only overall performance but a lack of ablation study and analysis. As a method that combines active learning and synthetic data generation from LLM, the authors only compare it with active learning approaches, and they should also compare the proposed method with synthetic data generation without active learning.

**Additional Comments On Reviewer Discussion:**

Reviewer aZDs raised concerns about the proposed method not generating the most informative data points and the niche settings of the paper.  The reviewer also pointed out that the experiments were limited and the improvements were hardly distinguishable.  Reviewer VPBQ asked for a detailed explanation of the state and action settings in the reinforcement learning scenario.  The reviewer also suggested comparing the proposed method with synthesizing-based methods.  Reviewer JtSL asked about the ratio of generated samples to actual unlabeled data queried at each iteration in the active learning process.  The reviewer also asked how the generated samples were human-labeled.

The authors responded to Reviewer aZDs by stating that they would add more experiments and ablation studies.  They also stated that they would clarify the settings of the paper.  The authors responded to Reviewer VPBQ by providing a detailed explanation of the state and action settings in the reinforcement learning scenario.  They also stated that they would compare the proposed method with synthesizing-based methods.  The authors responded to Reviewer JtSL by stating that they would add a definition of the text encoder model and an ethics statement.  They also stated that they would disclose the hyperparameter settings and add information about the human labeling process.

In my final decision, I weighed the points raised by the reviewers as follows:

I agreed with Reviewer aZDs that the proposed method does not generate the most informative data points and that the settings of the paper are a niche.  I also agreed that the experiments were limited and the improvements were hardly distinguishable.

I agreed with Reviewer VPBQ that a detailed explanation of the state and action settings in the reinforcement learning scenario was needed.  I also agreed that the proposed method should be compared with synthesizing-based methods.

I agreed with Reviewer JtSL that the ratio of generated samples to actual unlabeled data queried at each iteration in the active learning process was an important consideration.  I also agreed that the human labeling process needed to be described.

---

### Decision · Program_Chairs · 2025-01-22

Reject